# Transferability of Human and Environmental Microbiome on Clothes as a Tool for Forensic Investigations

**DOI:** 10.3390/genes15030375

**Published:** 2024-03-19

**Authors:** Noemi Procopio, Giulia Sguazzi, Emma V. Eriksson, Nengi Ogbanga, Frazer C. McKell, Eleanor P. Newton, Paola A. Magni, Andrea Bonicelli, Sarah Gino

**Affiliations:** 1School of Law and Policing, Research Centre for Field Archaeology and Forensic Taphonomy, University of Central Lancashire, Preston PR1 2HE, UK; nprocopio@uclan.ac.uk (N.P.); abonicelli@uclan.ac.uk (A.B.); 2CRIMEDIM—Center for Research and Training in Disaster Medicine, Humanitarian Aid and Global Health, Università del Piemonte Orientale, Via Lanino 1, 28100 Novara, Italy; giulia.sguazzi@uniupo.it; 3Department of Health Science, University of Piemonte Orientale, Via Solaroli 17, 28100 Novara, Italy; 4Department of Immunology, Genetics and Pathology, Uppsala University, Dag Hammarskjölds Väg 20, 75185 Uppsala, Sweden; emmaviolae@gmail.com; 5Forensic Science Research Group, Faculty of Health and Life Sciences, Applied Sciences, Northumbria University, Newcastle Upon Tyne NE1 8ST, UK; nengi.ogbanga@northumbria.ac.uk; 6School of Medical, Molecular & Forensic Sciences, Murdoch University, 90 South Street, Murdoch, WA 6150, Australia; frazer4muni@gmail.com (F.C.M.); 3illie@hotmail.co.nz (E.P.N.); p.magni@murdoch.edu.au (P.A.M.)

**Keywords:** skin microbiome, microbiome transfer, touch microbiome, personal identification

## Abstract

Considering the growing importance of microbiome analyses in forensics for identifying individuals, this study explores the transfer of the skin microbiome onto clothing, its persistence on fabrics over time, and its transferability from the environment and between different garments. Furthermore, this project compares three specific QIAGEN microbiome extraction kits to test their extraction efficiency on fabric samples. Additionally, this study aims to check if these extracts contain human DNA, providing a chance to obtain more information from the same evidence for personal identification. The results obtained show: (1) variations in the skin microbiome between the volunteers, potentially due to their different sex; (2) differences in microbial composition between worn and unworn clothing; (3) the influence of the environment on the microbial signature of unworn clothing; (4) the potential use of certain phyla as biomarkers to differentiate between worn and unworn garments, even over extended periods; (5) a tendency towards extraction biases in the QIAampMP^®^ DNA microbiome kit among the three tested ones; and (6) none of the extraction kits allow for the typing of human genetic profiles suitable for comparison. In conclusion, our study offers supplementary insights into the potential utility of time-transferred microbiome analysis on garments for forensic applications.

## 1. Introduction

Microorganisms are present ubiquitously on both the interior and exterior surfaces of the human body, outnumbering human cells in a ratio of 3:1 [1]. Similar to human DNA shed from skin cells, these microbes are also shed and transferred to the surrounding environment [2,3]. During the anthrax letter attacks in 2001 [4,5], the traditional investigative resources were inadequate, resulting in the impossibility of identifying the responsible party. Understanding the origins of a microorganism presented a unique challenge. These limitations gave rise to a novel area of study known as forensic microbiology. Originally, the Federal Bureau of Investigation defined this discipline as ‘dedicated to analysing evidence from a bioterrorism act, biocrime, or inadvertent microorganism/toxin release for attribution purposes’ [6]. Since then, the field has expanded and continued to develop into several areas and applications. Microbiome analysis has proven valuable in postmortem interval (PMI) estimations, and in establishing connections between a person or item and a geographical location (or crime scene) via soil [7], to name two of the most predominant applications.

The circular structure of bacterial DNA and the resilience of the peptidoglycan cell wall contribute to the enhanced durability of bacterial DNA, allowing it to withstand degradation much more effectively than human DNA. Low biomass samples are rarely an issue, and there is enough microbial DNA in a fingerprint to generate a reliable profile [2]. The specificity of a microbial profile, at strain-level taxonomic depth, is comparable to that of a fingerprint [8], making it highly individual. In addition, due to its high copy-number and high stability over time [9,10], a microbiome also represents a source of information that does not require a standard for comparisons [2]. Furthermore, bacterial profiles can be used independently of the environment and also under less than optimal circumstances, including, for example, difficult footprint analyses with smudged fingerprints or highly texturised surfaces [2,9]. With an estimated 500–1000 different bacterial species in the human body [8], it is not surprising that the genetic composition of the individual microbiome is even more diverse than that of human DNA, so that it can also be used to distinguish monozygotic twins [11]. Moreover, the microbial signature of a person can reveal more than human DNA might, providing insights into the person’s lifestyle and health status [3,12]. Given this potential, having the ability to analyse both nuclear DNA (nDNA) and microbial DNA from the same sample would facilitate the implementation of new investigative tools while still leveraging the tested and validated methods applied in forensic genetics. With specific reference to the skin microbiome, the literature reports its forensic uses for high-accuracy PMI estimation [13] and personal identification based on the microbial signature left on personal items, such as keyboards and computer mice [3,10]. Admittedly, a major limiting factor of using microbial profiles is that they are made of a community of living organisms, the structure and composition of which will change with alterations in their surroundings [8]. However, studies have demonstrated that the primary characteristics of the individual microbiome community at a specific body site are stable over time [3,10]. Although the skin microbiome is more diverse than that of internal body sites, certain core taxa constitute over 90% of the microbiome at a given site [3,14,15].

Based on the extensive data from the Human Microbiome Project, Huse et al. [15] tried to identify a core human microbiome. For the skin, some of the most prevalent taxa were Actinomycetales, at order-level taxonomic depth, and *Propionibacterium*, *Staphylococcus*, *Clostridium*, *Corynebacterium*, and *Mycobacterium* at genus level [15]. Schmedes et al. [10] also present *P. acnes* as a key organism. In a previous study conducted by Procopio et al. [3], Firmicutes, Proteobacteria, and Actinobacteria were the main phyla represented on human hands. Prominent taxa were also identified as part of a potential core ’touch microbiome’, as well as several potentially individualising biomarkers, suggesting the applicability of these analysis in a forensic context. Furthermore, they provided some evidence for the presence of ‘good’ and ‘bad’ microbiome shedders, similarly with the concept of good and bad human DNA shedders [3].

A theoretical scenario where forensic microbiome analyses may be useful could involve, for example, the case of an offender disposing of their clothes after committing a crime. According to Edmond Locard’s principle, when partaking in a criminal action, a potential perpetrator may leave traces behind but may also carry away traces from the crime scene, such as on clothing. In areas beyond the field of medicine, understanding the behaviours of bacteria remains largely elusive. It is known, however, that microbes, similar to nDNA, transfer onto touched surfaces and objects [3,9,12]. Although many studies conducted so far have focused on the microbiome of human palms or fingertips, clothing represent an item with which individuals come into close contact daily [16]. Given this, a logical progression in the pursuit of knowledge involves examining how skin microbiomes interact with clothing and whether this interaction can be used in a forensic context.

Due to the scarcity of microbiome studies simulating real forensic case scenarios [8] and their limitations due to the timescales selected [10] or to the surfaces taken under evaluation [9], this work represents a proof-of-concept study on the feasibility of microbiome analyses from tissues of forensic interest. Due to the limited sample size, it does not intend to provide quantitative or statistically relevant assessments of the reported findings. Specifically, the aims of this study are to clarify if microbes can transfer from individuals to their clothing and determine their persistence (up to six months), to investigate the transferability of the microbiome from the environment to clothing, to evaluate whether microbes transfer from a worn garment to an unworn garment adjacent to it, and to verify which of the kits marketed by QIAGEN (QIAamp^®^ PowerFecal^®^ Pro DNA, DNeasy^®^ PowerSoil^®^ Pro, and QIAamp^®^ DNA Microbiome) is the most suitable for the extraction of microbial and human genomic DNA from fabric.

## 2. Materials and Methods

### 2.1. Samples Collection

Two participants (from here on, ‘Individual 1’ (male) and ‘Individual 2’ (female)) of similar age and living in the same geographical area were provided with three sets of clothing (100% white cotton T-shirt) and personal protection equipment (PPE) to avoid contamination during sampling. All the new clothes used in the experiment were previously washed together at 60 °C with bleach and dried in a drier, then worn for 24 h at home. During this time, participants were required to engage in minimal activities, avoid meeting with other people, and adhere to the same meal plan. While lying on couches or beds for the duration of the experiment, participants were instructed to cover any furniture with a double cotton sheet, previously washed in the same manner as the clothing. After the 24 h, the worn garments were placed in individual cardboard boxes next to ones from the same lot and never worn. The same experiment was repeated by both participants three times on three different days to generate three sets of samples (‘1’, ‘2’, and ‘3’) each. Due to costs limitations, only subsets of the collected samples where then submitted for sequencing (Table 1 and Table 2).

Following the experiment, samples were cut from the neck of the T-shirt at regular intervals up to six months, and each one was then cut again to generate three technical replicates (‘a’, ‘b’, and ‘c’), subsequently used for the testing of the three different extraction kits. All samples were frozen at −20 °C until processing. Three positive controls were taken each day of the experiment for each participant from the skin of the back of the neck area before and after showering with QV soap by swabbing, using wet sterile cotton swabs with 0.09% NaCl for a total of 30 s. The swabs were subsequently frozen at −20 °C. QV soap is free from soap, colour, fragrance, propylene glycol, and lanolin, with a balanced pH.

### 2.2. Microbiome Extraction

Microbial DNA was isolated from the swabs and fabric samples using three different QIAGEN kits (Qiagen, Hilden, Germany): the DNeasy^®^ PowerSoil^®^ Pro Kit (replicates ‘a’), QIAamp^®^ PowerFecal^®^ Pro DNA (replicates ‘b’) Kit, and QIAamp^®^ DNA Microbiome Kit (replicates ‘c’) with a modified protocol. In total, 800 μL of Solution CD1 was added to Eppendorf tubes containing swab samples, vortexed for 30 s, and centrifuged at 5000 rpm for 10 min. After the fabric pieces were removed by pulling them through a tightly gripped sterilised tweezer to release as much of the liquid as possible, the supernatant was then transferred to the PowerBead Pro Plate, after which the extraction was performed according to the manufacturer’s protocol. For the QIAamp^®^ DNA Microbiome Kit (Qiagen, Hilden, Germany), 800 μL of Solution AHL was added to Eppendorf tubes containing swab samples, incubated at 70 °C, 600 rpm for 30 min. Also, in this case, the fabric pieces were pulled through a tightly gripped tweezer. Then, the extraction was performed according to the manufacturer’s protocol. DNA was eluted in 35 µL for tissue samples and 50 µL for positive control swabs. DNA quantification was performed with a NanoDrop One Microvolume UV-Vis Spectrophotometer (ThermoFisher Scientific, Waltham, MA, USA). As a quality check, amplification and gel electrophoresis of extracted microbial DNA was carried out. The samples and a positive control, in the form of *E. coli* TOP10 PLO3 D55N HOX A., were amplified using a Platinum™ Hot Start PCR Master Mix from Invitrogen™ (Waltham, MA, USA). A 1.5% (*w*/*v*) agarose gel made with 1× TAE buffer was used to check the quality of PCR products obtained prior to library preparation and sequencing.

### 2.3. Sequencing and Data Analysis

Library preparation and sequencing of the 16S rRNA gene (V4 region) performed on the Illumina MiSeq System with the wet-lab MiSeq SOP [17] was analysed by the NU-OMICS facility (Northumbria University, Newcastle, UK). PCR was carried out using the KA-PA2G Robust HotStart^®^ PCR Kit (from Sigma-Aldrich, St. Louis, MO, USA). PCR products were normalised using the Quant-iT™ PicoGreen™ dsDNA Assay Kit (Invitrogen™, Waltham, MA, USA) as described in the manufacturer’s instructions. Following the gold standards suggested by the Human Microbiome Project, targeting and sequencing of the V4 region of the 16S rRNA for bacterial identification were carried out. FASTq raw sequencing files that were generated were imported into QIIME2 software ver. 2021.11 [18] for analysis, quality filtered using DADA2 (QIIME2 DADA2 denoise-paired) [19], and trimmed at 235 bp forward and 220 bp reverse. Taxonomic assignments were achieved using silva-138-99 operation taxonomic units (OTUs) [20,21]. The processed data were then analysed in R (version 4.2.3), with the ‘Phyloseq’ packages [22]. Contaminants (archaea and fungi) and positive/negative controls were removed. PCoA plots based on Bray–Curtis dissimilarity and diversity bar plots were generated for a variety of subsets. Observed and Shannon diversity indices were calculated for different sample subsets and compared using an Analysis of Variance and *t*-test, with thhe significance set at α≤ 0.05. A linear discriminant analysis was performed with the ‘microbiomeMarker’ [23] R library with a Wilcoxon and Kruskal–Wallis cut-off set at α≤ 0.05. For a few sample subsets, the core microbiome profiles were generated. This was used to see the transfer between person and shirt, as well as worn shirt to unworn shirt.

### 2.4. Human STRs Profiles

To verify the existence of a possible dual use of extracts, STR profiles were typed from control swabs (M03_B_N_a, M03_A_N_a, M03_B_N_b, M03_A_N_b, M03_B_N_c, and M03_A_N_c). Following the extraction of human and microbial DNA, quantification was performed using the Qubit™ dsDNA HS Assay Kit and Qubit^®^2.0 (Invitrogen), following the manufacturer’s protocol. Amplifications were carried out using the GlobalFiler™ PCR Amplification Kit (ThermoFisher Scientific Waltham, MA, USA). Amplicons were run on a SeqStudio™ Genetic Analyzer (ThermoFisher Scientific Waltham, MA, USA), and finally, STR profiles were analysed using GeneMapper™ ID-X1.6 software.

## 3. Results

While the findings show promising trends, it is important to note that these results are preliminary, and further investigation is warranted for a comprehensive understanding of microbiome transfer to clothing.

### 3.1. Microbiome Transfer from Skin to Clothes

Observing the trends and progressions of microbial concentrations present on the tissue samples (clean T-shirts versus worn T-shirts), there is a noticeable increase of microbial presence in the garments worn by the ‘Individual 2’ participant and sampled at time zero in comparison with garments sampled directly after washing (Appendix A).

To evaluate the actual transfer of microbiome from human skin to the worn clothes, we compared the control T-shirts after their washing, the T-shirt worn from ‘Individual 1’ for 24 h and then sampled immediately after its removal (’T0’), and the neck swabs collected before and after a shower from ‘Individual 1’. A PCoA plot with Bray–Curtis dissimilarities (Figure 1A) shows more similarity between the swabs and the worn items than with the clean T-shirt control (Figure 1A) despite some differences in their composition (Figure 1B), with lower relative abundances of Proteobacteria in the swabs than the fabric samples and higher abundances of Basidiomycota, especially in the swabs taken before a shower, in comparison with the garments. Additionally, a clear separation between worn and unworn items (Figure 1A,B) can be noticed, where a greater abundance of specific phyla (such as Firmicutes) is found in the worn garment in comparison with the unworn one.

### 3.2. Stability of the Microbiome Transferred on Clothes and Transferability between Adjacent Garments

To evaluate the stability of the transferred microbiome on garments, we initially looked at the DNA quantification results. DNA quantification of the ‘Individual 2’ samples collected after increasing time points after wearing the garment (T1, 2, 3 weeks and T2 months) shows no clear increase or drop in DNA concentrations over time (Appendix A). PCoAs based on Bray-Curtis dissimilarities performed on both ‘Individual 1’ (up to 180 days) and ‘Individual 2’ (up to 60 days) samples show no detectable pattern related to time. Instead, samples cluster clearly with respect to the participant (Figure 2A) or with respect to the condition of the item (worn vs. unworn) (Figure 2B). It is evident that the strongest clusters are found for the worn samples of each participant, whereas the samples collected from the unworn items are not as densely clustered, suggesting a lower degree of within-group similarity than for the worn items. Temporal stability within the person wearing the clothing for different days was also examined (‘Individual 2’ sample sets ‘1’, ‘2’, and ‘3’ of the replicated). For different days, the microbiome transferred to the worn shirts seems to be, at large, interchangeable. Specifically, the worn shirts from sets ‘1’ and ‘3’ cluster very close to each other, and the worn shirts from set ‘2’, despite clustering sometimes more far away from the other two, are still closer to the other ’worn’ items than to any ’unworn’ one (Figure 2).

When considering the alpha diversity and the abundances of specific phyla over time (Figure 3), it is noticeable that there is a similar lack of a time-related pattern, especially for the ‘Individual 1’ samples (Figure 3A,B). ‘Individual 2’ samples show a decrease in alpha diversity from T0 to increasing time points, but also in this case, besides T0, there is a lack of correlation between times and microbial diversity (Figure 3C,D). The most abundant phylum for unworn items is Proteobacteria both in the ‘Individual 1’ and in the ‘Individual 2’ samples, whereas Firmicutes became the most abundant phylum in worn ‘Individual 1’ samples and Actinobacteriota and Firmicutes become abundant in worn ‘Individual 2’ samples in comparison with the unworn items. A linear discriminant analysis (LDA) of the experimental samples collected from ‘Individual 1’ (Figure 4A) and ‘Individual 2’ (Figure 4B) participants confirms which are the most significant taxa, which may be used as biomarkers to discriminate if an item was worn or not. Despite minor differences between the ‘Individual 1’ and the ‘Individual 2’ samples, members of the Firmicutes phylum (such as Lactobacillales (Streptococacceae family), Staphylococcales (Staphylococcaceae family), Veillonellales–Selenomonadales (Veillonellaceae family), and Clostridiales (Peptostreptococcaceae family)), of the Actinobacteria phylum (Corynebacteriales bacteriales—Corynebacteriaceae family), of the Proteobacterium phylum (Neisseriales—Neisseriaceae family), of the Actinomycetota phylum (Actinomycetales—Actinomycetaceae family), and of the Bacteroidota phylum (Bacteroidales—Prevotellaceae family) may be used as biomarkers to discriminate if an item was worn or not.

By looking more in detail at the core microbiome found on both worn and unworn items after excluding the microbiome found after washing the garments for both participants, we identified specific ASVs able to be transferred from the worn to the unworn garments and remain stable over time, up to 180 days (Appendix A). Specifically, we identified a *Streptococcus* sp. (unique ASV code 06f825b512d903b9230e1a55d87359ee) and a *Micrococcus* sp. (unique ASV code 0c579d21280801f02a641e1608606927) in both worn and unworn items from 14 days up to 180. Of them, only the *Streptococcus* sp. was also identified as part of the core microbiome of the ‘Individual 1’ swab samples (before and after shower), whereas the *Micrococcus* sp. did not belong to the core microbiome of the swabs. Interestingly, these species were not found in the box swabs collected on the inner surface of the box where the garments were placed (prior to their deposition) either. Three of the four core ASVs identified in the garments post-washing were also found on the swabs (*Staphylococcus* genus, 65d43491988bfe557da4d86a5ba25dae, *Corynebacterium* genus, aa9b3a1418d146c262ec63305292065a, and *Corynebacterium aurimucosum*, 6a4c0e5943a7eb8c9f0b5b5e69171828), whereas a unique one (*Acinetobacter* genus, ea403646ed22d679fa4586263d8fc32f) was found specifically on the samples acquired after washing the garments.

### 3.3. Evaluation on Microbial Transferability and Inter-Individual Variations

As previously highlighted, a significant difference was found between the microbiome of the two participants both on the worn and, consecutively, on the unworn items. When focusing exclusively on the unworn garments, including the control samples obtained prior to the beginning of the experiments, we identified the presence of specific clusters related with the two participants (Figure 5A). Interestingly, this was also observed for the control samples taken on the washed items prior to the beginning of the experiment. The samples collected by the ‘Individual 2’ participant showed higher abundances of Proteobacteria than the ‘Individual 1’ ones, and a lower abundance of Actinobacteriota and Firmicutes (Figure 5B). These differences may not necessarily be linked with the biological sex of the participants, as the limited sample size does not allow us to reach this conclusion, and more research is required to specifically address the topic of transferability in association with biological sex differences.

### 3.4. Comparison of Efficiency between Three Extraction Kits

To understand how different kits perform when extracting microbial DNA from fabric, three QIAGEN kits were tested on the three technical replicates collected from the ‘Individual 1’ participant at each time point. The results from the quantifications of the extracts (Appendix A) do not show any distinct pattern. The only consistent detail of note is the slightly lower yield obtained with the QIAamp^®^DNA Microbiome Kit. Also, in terms of observed alpha diversity and Shannon-indexed alpha diversity, the differences between the three kits results in being non-significant. Observing the phylum relative abundances extracted from the three technical replicates (Figure 6), a clear difference is noticeable, particularly for the worn samples, in the results obtained with the QIAamp^®^DNA Microbiome Kit in comparison with the others. In this case, samples at different time points tend to be quite different in their microbial composition, whereas the DNeasy^®^PowerSoil^®^Pro Kit shows more stable distributions across time, similar to the QIAamp^®^PowerFecal^®^Pro DNA Kit, where only T0 samples are distinct from all the remaining ones. Specifically, the QIAamp^®^DNA Microbiome Kit shows a tendency of extraction bias, displaying higher proportions of Firmicutes and lower proportions of, e.g., Proteobacteria and Bacteroidota, along with other taxa represented at a lower degree.

Observed and Shannon alpha diversity indices do not show statistical differences between the various kits (Figure 7A); however, it is clear from the PCoA that samples extracted with the QIAamp^®^DNA Microbiome Kit are distinguishable from the samples extracted with the other two kits, especially for the worn items (Figure 7B), confirming the observations made on their phylum distributions.

### 3.5. Human STR Profile

An aliquot of the six extractions, obtained from the ‘Individual 1’ skin swabs (M03_B_N_a, M03_A_N_a, M03_B_N_b, M03_A_N_b, M03_B_N_c, and M03_A_N_c) collected before and after a shower was amplified for 24 human STRs. Three out of the six samples resulted in incomplete genetic profiles (Appendix A). The other three extracts yielded no alleles at any loci. Given the negative results of skin swab typing, we decided not to proceed with the tissue-based typing of human DNA.

## 4. Discussion

The notion that clothing items serve as reservoirs for commensal and pathogenic bacteria and fungi has received attention in the past two decades [24,25,26]. This consideration also extends to the potential transfer of bacteria and fungi from clothing to washing machine surfaces [27,28] and vice versa [24]. Knowledge of the abundance and species of microbes transferred in these processes becomes necessary in determining possible health implications, as well as determining strategies to mitigate any risks but leaving aside the potential utility in the forensic field. As Locard’s exchange principle states, the perpetrator of a crime leaves something at the scene and takes something with them. In this study, the aim was to apply this principle to the microbiome left on clothing [29] by evaluating the transfer from the skin of the neck of volunteers to T-shirts. For this experiment, only the shirt collar samples were analysed from the sampled clothing. This area was chosen because it has a high degree of contact between the skin and the fabric, increasing the chance of successfully studying the transfer. However, it was considered that previous studies have reported more significant microbial growth in moist environments, such as socks and underwear, compared to the drier and cooler environments provided by T-shirts [24]. Certainly, desiring this application in a potential forensic scenario, a shirt is more frequently encountered as an item often ’discarded’ after a crime compared to socks or underwear.

The primary interaction between clothing and skin is mechanical through friction and pressure and depending on factors such as fibre and fabric structure, material, and textile quality that may potentially favour some bacteria species over others [30,31]. In the literature, the potential transfer is not considered to be one-sided; rather, the fabric can also affect the skin microbiome, either by alternating its structure or by compromising its skin barrier function as a result of the pH elevation due to the occlusion of the skin by the garment itself [31,32]. Sanders et al. [33] believe that such a transfer may favour the selective bacterial enrichment of some microbes over others, promoting the creation of a new microbiome within the tissue, the composition of which may differ from that of the original skin microbiome. In summary, the authors describe this bacterial enrichment on tissues as depending largely on the bacterial community in the skin as well as on the bacterial species themselves, the contact conditions, and the physicochemical properties of the clothing’s fabric [31].

Phylum relative abundances, Bray–Curtis distances and core microbiomes identified between the clean control samples, and the swabs from the neck and the worn T0 shirt sample from ‘Individual 1’ have been compared to determine whether the transfer of the microbiome from the neck of a person to the shirt worn could be established with certainty. Our results, in agreement with the literature, show that the skin microbiome as well as the textile microbiome are dominated by the three phyla Firmicutes, Actinobacteria, and Proteobacteria [25,31,33] that assume different relative abundances depending on whether the T-shirt was worn or not.

Surprisingly, the analysis of the clean shirt after washing showed a similar microbial composition, in terms of phylum, to the worn T-shirt but with different relative abundances. Plausible explanations of that overlap are the contamination of the clean shirt by the participant during the manipulation of the samples or from the domestic environment itself [27,28,34]. Yet, worn and unworn T-shirts can be easily distinguished between the two when comparing one to the other. Given the ubiquitous nature of bacteria, finding a completely sterile environment is unlikely, especially if we consider the domestic one. In our study, the aim was to simulate a potential real-life scenario. Therefore, we decided not to conduct the experiment in ‘cleaner’ environments (e.g., laboratory setting).

Despite washing the garments with bleach at a high temperature, it was possible to identify microbial contamination on the clean items. As is well known, increased temperatures and the presence of additives such as bleach during the washing cycle reduce the bacterial biomass of garments and the potential for cross-contamination [34]. Nevertheless, as previously stated by Gattlen et al. [27], detergents containing bleach are known to reduce the microbial load in the washing machine, but, in particular, that deposited on the tissues and not on the biofilm formed on the mechanical components of the apparatus that may persist and must be considered as a possible source of contamination. Interestingly, all the clean items sampled by both volunteers were washed together in the same washing machine and packed individually before being brought to each volunteer’s home and being sampled by the volunteers. In the literature, it is well known that not only there are indigenous bacterial communities on unwashed and unworn cotton T-shirts, but also that there is a difference depending on the type of fabric [25]. In our study, where we tested only one type of fabric, the ’contamination’ present on the clean garments showed up different between ’Individual 1’ and ‘Individual 2’ samples and remained well distinguished throughout the entire experiment (e.g., also with non-worn items located in close proximity with the worn ones). The fact that the T-shirts sampled prior to the starting of the experiment show a distinct microbiome (‘Individual 1’ vs. ‘Individual 2’) reveals a potential contamination caused either by the participants when conducting the samples or by the environment (domestic) where samples were conducted. Therefore, the difference observed between ‘Individual 1’ and ‘Individual 2’ unworn items at various time points is likely to be associated with some form of contamination (as previously reported) and not necessarily with the microbial transfer between worn and unworn items. More recently, Yan et al. [35] pointed out that the microbiomes of clothes are mainly shaped by the individual wearing them and from the air particles carrying microorganisms differently adsorbed on fabrics [36].

In relation to the temporal stability of the transferred microbiome on the worn clothing, the results did not show significant microbial alternations associated with increasing time points, highlighting temporal stability both within and between samples. In principle, this suggests that the microbiome transferred from the skin to the garments can be sampled with confidence for at least up to six months after transfer. The temporal stability of the microbiome emerges as a key factor in the forensic domain. If the microbial composition of worn clothing undergoes temporal fluctuations, then the derivation of conclusive insights from a momentary community profile would be precluded. Our findings agree with Oh et al. [37], where it was shown that despite the continuous disruption that human skin undergoes in daily life, healthy adults permanently maintain their skin communities for up to two years, similar to the stability observed in the gut. The homeostasis that characterises the microbial communities in the skin is largely due to the fixation of abundant species, although smaller numbers of sporadic species are also permanent and contribute to an individual’s unique microbial signature [37]. Furthermore, we can conclude that in our study, there was not only a temporal stability in the transferred microbiome, but also within the person wearing the garments over the course of the three different days when the replicate experiments were conducted. This result opens the possibility of complementing STR analyses with the microbial signature of an individual for identification purposes.

With this study, we were able to identify potential biomarkers for the discrimination of worn versus unworn items. In fact, both in ‘Individual 1’ and ‘Individual 2’ samples, it was possible to associate the presence of specific taxa normally found on skin and mucosae such as Lactobacillales (*Streptococcus* sp.) [38], Staphylococcales [39], Veillonellales–Selenomonadales [40], Actinomycetales, Corynebacteriales [39], and Bacteroidota with worn items. Specifically, *Streptococcus* sp. also belonged to part of the core microbiome of both worn and unworn items and was also shared with the human swabs collected in the study. This is in contrast with *Micrococcus* sp., found to belong to the core microbiome of the garments but not identified on the neck swabs. *Micrococcus* sp., including, for example, *Micrococcus luteus*, is widely present in natural environments, such as soil and water resources, but also in dust and in the air. However, it is also considered a normal inhabitant of human skin and oropharynx mucosa [41,42]. The fact that this specific microbial genus was not found on the swabs suggests that its presence on the garments may be a consequence of environmental contamination and should not, therefore, be associated with the action of wearing the garments. Additionally, Clostridiales, found in the gut [43], in the environment [44], but also on human skin [45], were identified as potential indicators for worn items, similar to Neisseriales. Neisseriales, belonging to the Neisseriaceae family as well as Moraxellaceae, are found in water and soil; in contrast with Neisseriales, Moraxellaceae were found to characterise specifically the unworn items. Bacteria belonging to this genus are known to have a high occurrence on cotton fabrics [46]. Recent studies such as those of Calleweart et al. [47] showed that Moraxellaceae are found on the human body in the axillary region and are retained as responsible for malodour, different from what was previously reported by Kubota et al. [48]. The unworn clothing, including those collected prior to the start of the experiment (e.g., only washed) was characterised by Proteobacteria phylum as the most abundant one, both in ‘Individual 1’ and in ‘Individual 2’ samples. Proteobacteria and Bacteroidetes are known for their dominant presence in drinking water supplies [49]. However, Callewaert et al. [46] showed that Proteobacteria present in the influent water returned in high amounts in the effluent water, whilst only being marginally present in the cotton samples after laundering. Although the species identified in the worn samples included members of the commensal skin flora, such as Corynobacteriaceae and Peptostreptococcales–Tissierellales, species that are opportunistic pathogens, such as Staphylococcaceae and Streptococcaceae that can cause skin and soft-tissue infections, were also recovered [24]. The only bacterium found exclusively on the samples collected after washing was Acinetobacter guillouiae, an environmental gram-negative bacterium previously isolated from gasworks effluents [50,51].

The results also show clustering by gender. Despite this, these findings should be interpreted cautiously, as only two volunteers (one per gender) were included in the study. A significant difference was found in the relative abundances of the microbiome of the two participants between both worn and unworn garments. According to the literature [52,53,54], the ‘Individual 2’ skin microbiome is characterised by a greater species diversity than the ‘Individual 1’ microbiome. Presumably, such greater diversification of the ‘Individual 2’ skin microbiome results from the presence of thinner skin, lower pH, and less sweat production than in the ‘Individual 1’ skin [53,54,55]. As early as 2011, Grice and Segre wondered whether these differences were the outcome of physiological factors or of different hygiene and cosmetic habits [52]. Most recently, Skowron et al. [53] suggested that these differences in the microbiome composition between male and female result from sex-specific properties of the skin (the skin thickness, the number of hairs, sweat and sebaceous glands), sex hormones [56], daily personal hygiene, and from the use of cosmetics, of which the ingredients may promote or inhibit the growth of certain bacteria [53]. It is, therefore, conceivable that the difference observed here between ‘Individual 1’ and ‘Individual 2’ samples, especially for the worn items, may be linked to these biological differences previously observed between the two genders. However, it cannot be excluded that other inter-individual variations (not linked to the biological sex of the participants) may have also caused the observed separation between the samples.

Although the cutaneous microbiome for both sexes is dominated by four bacterial phyla: Actinobacteria, Firmicutes, Proteobacteria, and Bacteroidotes [57], the results obtained confirm what is reported in the literature on the greater bacterial diversity that characterises the ‘Individual 2’ cutaneous microbiome. Specifically, our results show a predominance of Proteobacteria in ‘Individual 2’ samples against a lower relative abundance of Actinobacteroidota and Firmicutes than ‘Individual 1’ samples that agree with that found by Robert et al. [58]. Aware of the limited volunteer cohort under investigation, it may be rash to venture a hypothesis; however, changes in skin pH may be influential. Men generally have more acidic skin than women, and this results in an often lower microbial diversity [59]. Alongside this, Firmicutes and Actinibacteria are known to be more tolerant to acidic pH [60].

When comparing the efficiency of the different extraction kits, only minor and non-significant differences were found. The most abundant taxa were shared, in fact, amongst all the kits tested. Furthermore, a lower yield is evident with regard to the QIAamp^®^DNA Microbiome Kit, which also shows differences in the relative abundances of the identified taxa in the three technical replicates collected from the worn samples. In addition, it is interesting to note that the QIAamp^®^DNA Microbiome Kit appeared to be more effective on the extraction of gram-positive bacteria (Firmicutes) than gram-negative ones (Proteobacteria and Bacteroidota), contrary to the theory that without a mechanical lysis step, the cell wall of gram-positive species would remain intact. As no kit was designed for the extraction of the microbiome from fabric, it was necessary to adapt the protocols, as described in the Section 2. However, the choice of the kit did not seem to have a significant impact on the results. The choice of using different extraction kits was also performed to test if, in addition to microbial DNA, any of these kits could also allow for the co-extraction of human DNA of suitable quality to be used for forensic identification purposes. The results obtained are not encouraging, as only three out of six samples gave an incomplete genetic profile. Previously, it was shown by Sguazzi et al. [61] that it is possible to obtain full microbial signatures from DNA extracts obtained with forensic kits specifically designed for human DNA. The fact that the analysis of the STRs failed in this instance is likely associated with the fact that these microbial kits are not optimised to maximise the obtainment of STR profiles from sub-ideal samples (e.g., skin swabs).

### Limitations

The authors are aware of several limitations of this study. (1) We only analysed the bacterial composition on cotton fabric and did not evaluate the transfer and persistence of bacteria on other fibres, such as polyester and mixed fabrics. (2) Although the measurements were repeated during the study, they originated from a limited number of volunteers (two). (3) Although the impact of environmental contamination was contained through the use of PPE, environmental samples in addition to the ones collected from the boxes should have been collected, such as those from the washing machine or those from the couch with which the subjects came into contact. (4) The presence of a potential ‘native’ microbiome on the fabric and how this might affect humans is not clear and was not assessed in this study. Further studies will need to clarify all these questions.

## 5. Conclusions

Despite the highlighted limitations, this study provides useful insights on the transferability of the human and environmental microbiome on clothing and on their persistence, and sets the bases for additional analyses in this direction. The fact that the transferred microbiome persists for prolonged periods of time without noticeable variations on garments supports the future application of this methodology to forensic settings and strengthens the interchangeability of extraction kits from tissue fibres that employ similar lysis methods.

## Figures and Tables

**Figure 1 genes-15-00375-f001:**
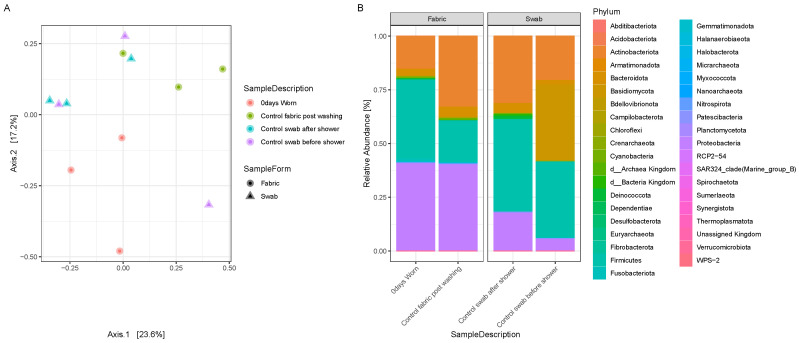
(**A**) PCoA plots displaying Bray–Curtis distances between a subset of samples, including a control T-shirt post-washing, the T-shirt worn by ‘Individual 1’ for 24 h and sampled immediately after removal (‘T0’), and the neck swabs before and after a shower; (**B**) bar plot displaying the relative abundances of phyla present in the control T-shirt, in the ’Individual 1’ worn T-shirt at T0, and in the control neck swabs before and after a shower.

**Figure 2 genes-15-00375-f002:**
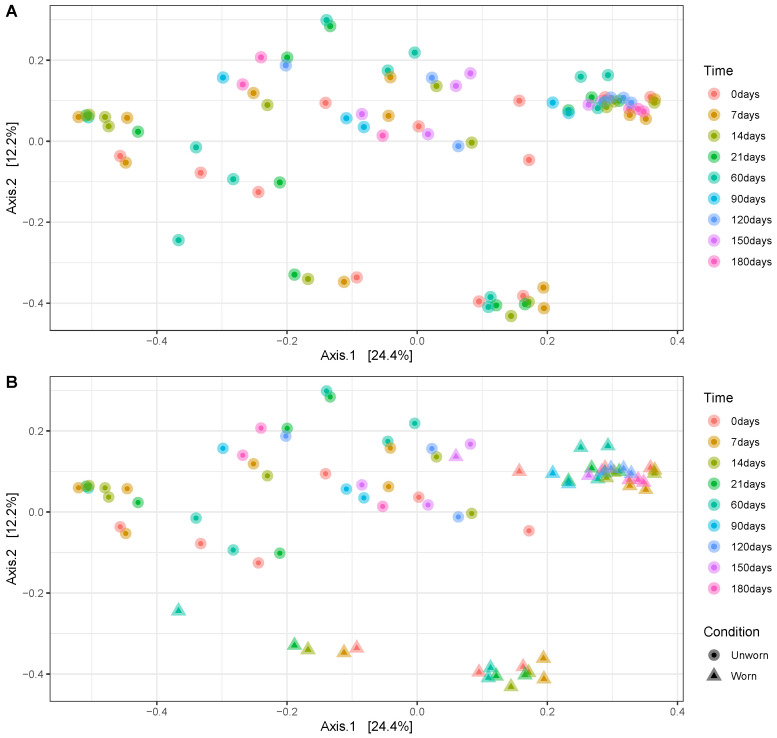
PCoA plot of the experimental samples during the whole duration of the experiment: (**A**) plot considering time (different colours) and individuals (different shapes); (**B**) plot considering time (different colours) and condition (different shapes).

**Figure 3 genes-15-00375-f003:**
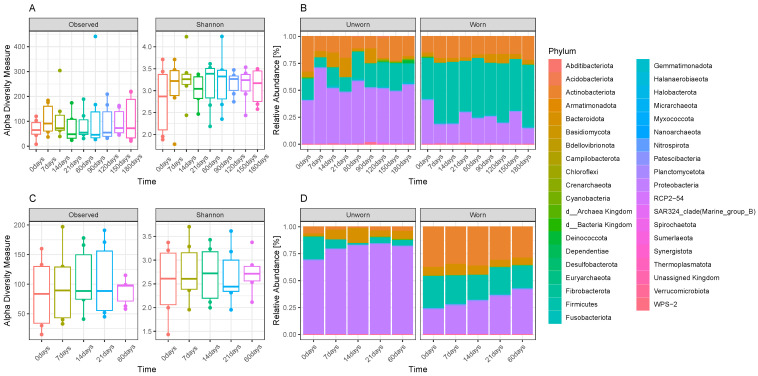
(**A**) Observed alpha diversity and Shannon diversity of the ‘Individual 1’ shirt samples over time; (**B**) bar plot displaying the average relative abundances of phyla present in the three sets (‘a’, ‘b’, and ‘c’) of ‘Individual 1’ samples over time, divided between worn and unworn items; (**C**) observed alpha diversity and Shannon diversity of the ‘Individual 2’ shirt samples over time; (**D**) bar plot displaying the relative abundances of phyla present in the ‘Individual 2’ samples over time, divided between worn and unworn items.

**Figure 4 genes-15-00375-f004:**
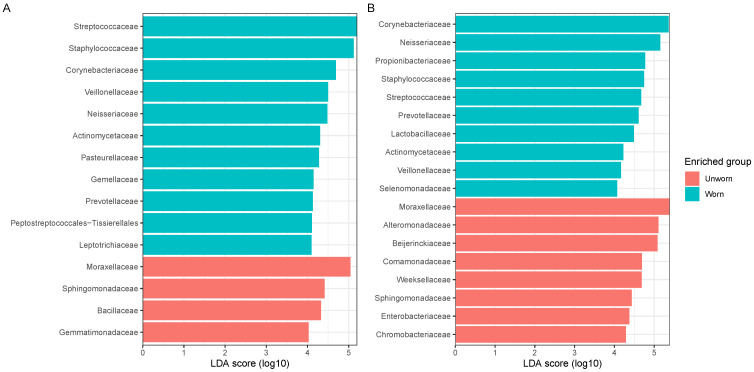
Linear discriminant analysis conducted at the family level for (**A**) ‘Individual 1’ samples and (**B**) ‘Individual 2’ samples for both worn and unworn items. The LDA score is represented in log10; only orders with a score > 4 are reported.

**Figure 5 genes-15-00375-f005:**
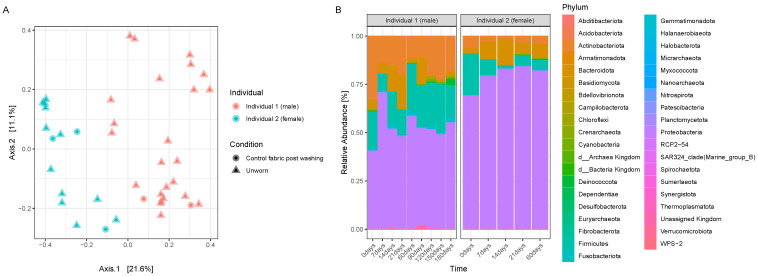
(**A**) PCoA of unworn items comparing ‘Individual 2’ versus ‘Individual 1’ samples, and (**B**) bar plot at the phylum level of unworn items, spanning 60 days for the ‘Individual 2’ samples and 180 days for the ‘Individual 1’ ones.

**Figure 6 genes-15-00375-f006:**
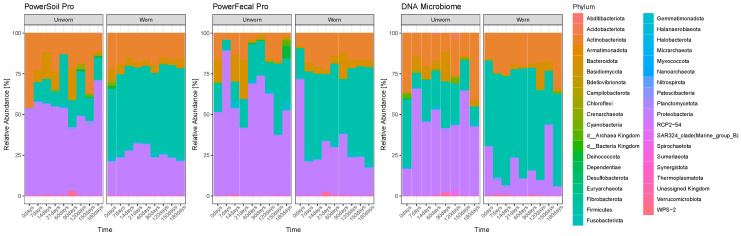
Bar plots with microbial relative abundances in both worn and unworn samples collected from the ‘Individual 1’ participant at the phylum level, separated by extraction kit.

**Figure 7 genes-15-00375-f007:**
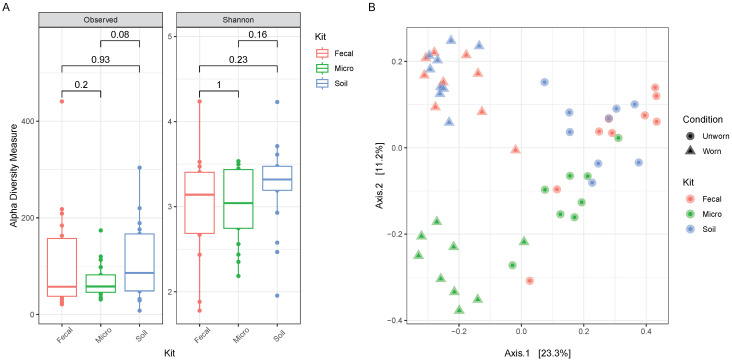
(**A**) Observed and Shannon-indexed alpha diversity of the experimental samples collected from ‘Individual 1’, based on the extraction kit used. Pairwise *t*-test *p*-values are reported numerically in the plots, and significance was set at α = 0.05. (**B**) PCoA of the samples collected from ‘Individual 1’, comparing the factor ‘worn’ versus ‘unworn’ and the kit used for extraction.

**Table 1 genes-15-00375-t001:** The ‘Individual 1’ (‘M’) samples sequenced in this project. The labels for the positive controls from the neck, represent the sampling before (‘B’) and after (‘A’) shower. The number ‘3’ after the letter ‘M’ indicates the third of a triplicate set; ‘SN’ is an abbreviation for ‘Shirt Neck’, and the difference between W and Q is that the W-labelled samples are those that were worn by the subject, while Q indicates unworn ones.

Sample Name	Time	Label	From
M03_B_N	Week 0	Control	Swab
M03_A_N	Week 0	Control	Swab
M03_SN	Week 0	Control	Fabric
M3_SN_W0_W	Week 0	Worn	Fabric
M3_SN_W1_W	Week 1	Worn	Fabric
M3_SN_W1_Q	Week 1	Unworn	Fabric
M3_SN_W2_W	Week 2	Worn	Fabric
M3_SN_W2_Q	Week 2	Unworn	Fabric
M3_SN_W3_W	Week 3	Worn	Fabric
M3_SN_W3_Q	Week 3	Unworn	Fabric
M3_SN_M2_W	Month 2	Worn	Fabric
M3_SN_M2_Q	Month 2	Unworn	Fabric
M3_SN_M3_W	Month 3	Worn	Fabric
M3_SN_M3_Q	Month 3	Unworn	Fabric
M3_SN_M4_W	Month 4	Worn	Fabric
M3_SN_M4_Q	Month 4	Unworn	Fabric
M3_SN_M5_W	Month 5	Worn	Fabric
M3_SN_M5_Q	Month 5	Unworn	Fabric
M3_SN_M6_W	Month 6	Worn	Fabric
M3_SN_M6_Q	Month 6	Unworn	Fabric

**Table 2 genes-15-00375-t002:** The ‘Individual 2’ (‘F’) samples sequenced in this project. The number ‘1’ after the letter ’F’ indicates the first of a triplicate set; ‘SN’ is an abbreviation for ’Shirt Neck’, and the difference between W and Q is that the W-labelled samples are those that were worn by the subject, while Q indicates the ones unworn. N.B. The ’SN’ control was taken by using a swab and not by directly cutting the fabric.

Sample Name	Time	Label	From
F01_SN	0	N/A	Swab
F1_SN_W0_W	0	Worn	Fabric
F1_SN_W1_W	Week 1	Worn	Fabric
F1_SN_W1_Q	Week 1	Unworn	Fabric
F1_SN_W2_W	Week 2	Worn	Fabric
F1_SN_W2_Q	Week 2	Unworn	Fabric
F1_SN_W3_W	Week 3	Worn	Fabric
F1_SN_W3_Q	Week 3	Unworn	Fabric
F1_SN_M2_W	Month 2	Worn	Fabric
F1_SN_M2_Q	Month 2	Unworn	Fabric

## Data Availability

The data presented in this study are available on request from the corresponding author. The data are not publicly available due to privacy.

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
