# Peer review of "Transferability of Human and Environmental Microbiome on Clothes as a Tool for Forensic Investigations"

_genes, 2024, doi:10.3390/genes15030375_

Round 1

Reviewer 1 Report

Comments and Suggestions for Authors

Review on the manuscript of Procopio N et al.: “Transferability of human and environmental microbiome on clothes as a tool for forensic investigations”.

In this study, authors explored the transfer of the skin microbiome onto clothing, its persistence over time, and its transferability from the environment and between different garments. In addition, Authors tested whether tested the extraction efficiency of three specific QIAGEN microbiome extraction kits. Authors observed differences in microbial composition between worn and unworn clothing, that certain phyla may have potential to be used as biomarkers to differentiate between worn and unworn garments, even over ex-tended periods, that all kits tested show similar extraction efficiencies and that none of the extraction kits allowed the typing of human genetic profiles suitable for comparison.

Overall, I felt that this topic has some interest. Therefore, the elucidation of a putative transfer of human microbiota to garments may have potential for forensic investigations. The manuscript is clear and well organized. In addition, I consider that the manuscript is precise on the questions that Authors proposed to study. Thus, the issues that arise to me are listed below, so, I hope Authors find the following comments and suggestions useful.

1 - Two participants of different sexes were enrolled in the study. Do Authors consider that two participants are enough to take valid conclusions from the study?

2 - In line 194, please correct “Figure ??B”.

3 - What factors were considered when selecting the shirt collar for analysis?

4 - Authors saw that samples collected from the female participant showed a higher abundance of Proteobacteria than the male, but lower abundance of Actinobacteriota and Firmicutes. Do Authors can formulate a hypothesis for such results and explore it in the discussion section?

5 - In the discussion section, Authors indicate that “a shirt is more frequently encountered as an item often “discarded” after a crime, compared to socks or underwear”. Based on this, can Authors justify why shirts were used for analysis, instead of socks or underwear?

6 - According to the literature cited, what factors contribute to differences in the microbiome composition between males and females?

Comments on the Quality of English Language

Some minor mistakes detected.

Author Response

Dear Reviewer,

in the attached file you will find our answers to your comments/suggestions. We hope we have answered clearly and comprehensively.

Reviewer 2 Report

Comments and Suggestions for Authors

This study investigates the transfer of skin microbiome onto clothing, its persistence on fabrics, and its transferability between garments and the environment. It also compares the efficiency of three QIAGEN microbiome extraction kits on fabric samples and examines the presence of human DNA in the extracts for personal identification. Results reveal variations in skin microbiome between volunteers, differences in microbial composition between worn and unworn clothing, and the impact of the environment on microbial signatures. Certain phyla could potentially serve as biomarkers to differentiate between worn and unworn garments, even over extended periods. However, none of the extraction kits allowed for the typing of human genetic profiles suitable for comparison. Overall, this study provides additional insights into the potential utility of time-transferred microbiome analysis on garments for forensic applications.

Comment:

The study's limited sample size of only two participants restricts the generalizability of its findings. Additionally, the method of participant selection is not specified, potentially introducing bias. Given the importance of this research, the authors should consider conducting a larger study with a more diverse participant pool. 

Author Response

Dear Reviewer,

in the attached file you will find our answer to your comment. We hope we have answered clearly and comprehensively.

Round 2

Reviewer 1 Report

Comments and Suggestions for Authors

Second review on the manuscript of Procopio N et al.: “Transferability of human and environmental microbiome on clothes as a tool for forensic investigations”.

In this study, authors explored the transfer of the skin microbiome onto clothing, its persistence over time, and its transferability from the environment and between different garments. In addition, Authors tested whether tested the extraction efficiency of three specific QIAGEN microbiome extraction kits. Authors observed differences in microbial composition between worn and unworn clothing, that certain phyla may have potential to be used as biomarkers to differentiate between worn and unworn garments, even over ex-tended periods, that all kits tested show similar extraction efficiencies and that none of the extraction kits allowed the typing of human genetic profiles suitable for comparison.

This represents a second version of the manuscript after peer-review. Authors have clarified all questions raised by me in the first revision round. So, I congratulate Authors for the clear explanations. Meanwhile, although Authors claim that this represents a preliminary study, in an ideal scenario, some additional, more consolidated data should be included, which would give more relevance to the study. However, I agree with the Authors’ point-of-view that this represents just a preliminary study.

Reviewer 2 Report

Comments and Suggestions for Authors

The authors have satisfactorily addressed the concerns raised in the previous report. As proposed it would be beneficial to implement the information obtained through this study to a larger sample size and get more comprehensive data in the future.